# Ovarian torsion: A retrospective case series at a tertiary care center emergency department

**Faysal Tabbara**[1], **Moustafa Hariri**[2], **Eveline Hitti**[1]*

**1** Department of Emergency Medicine, American University of Beirut Medical Center, Beirut, Lebanon, **2** QU Health, Vice President for Medical and Health Sciences, Qatar University, Doha, Qatar

* eh16@aub.edu.lb

**Data Availability Statement:** All relevant data are within the manuscript and its Supporting Information files.

## Abstract

Ovarian torsion (OT) is a gynecological emergency that can have diverse clinical presentations makings its diagnosis especially challenging. Few studies exist in the literature describing the clinical presentation as well as the management of OT in the emergency department (ED). This study aims to describe the clinical presentation, physical examination, emergency management, radiographic and intraoperative findings, histopathology reports and the time-to-intervention metrics of OT cases presenting to the emergency room of our tertiary care center. Data was collected by a retrospective chart review of all OT cases that presented to our ED over a period of 1 year. A total of 20 cases were included in the final analysis. The incidence of OT in the ED was 157.4 per 100 000 visits of women in the reproductive age group. All patients were women of reproductive age, with a mean age of 27.3 years. A total of 15 patients (78.9%) presented within 24 hours of symptom onset. The most common presenting symptom was abdominal pain (95%, 19/20) with most localizing to the right lower quadrant (60%, 12/20), followed by nausea and vomiting (90%, 18/20). Almost all patients (95%, 19/20) required opioids for pain management. The majority (80%, 16/20) were diagnosed by ultrasound prior to surgery, whereas (20%, 4/20) went straight to surgery based on a high index of clinical suspicion. Among those who underwent ultrasound, ovarian cyst was present in (75%, 12/16) while (68.9%, 11/16) showed ovarian enlargement and (50%, 8/16) showed abnormal ovarian blood flow. All patients except one patient underwent operative management. Out of the 19 patients who underwent surgery, the majority of patients (94.7%, 18/19) had viable ovaries with the exception of 1 patient (5.3%) who required a salpingo-oophorectomy post ovarian detorsion. A total of 13 cases included histopathologic review of specimens out of which 6 (46.2%) had a mature cystic teratoma. The mean time from door to ultrasonography was 1.4 hours and from door to surgery was 11.4 hours. Our study provides valuable insights into the incidence, clinical presentation, physical examination, emergency management, ultrasonographic and intraoperative findings, histopathology reports as well as the time-to-intervention metrics of OT patients presenting to the ED.

**Funding:** The author(s) received no specific funding for this work.

**Competing interests:** The authors have declared that no competing interests exist.

## Background

Ovarian torsion (OT) is a critical gynecological condition which involves the rotation of the ovary around its pedicle, impairing blood flow and raising the risk of ischemic injury. Mostly affecting women of reproductive age, it is a rare yet serious condition as a missed or delayed diagnosis is associated with impaired or lost fertility [1]. Its etiology is usually multifactorial, but it is commonly associated with underlying ovarian masses or cysts, making the ovary more susceptible to torsion. While patients with gynecologic emergencies commonly present to the ED, ovarian torsion represents only 3% of gynecological emergencies [2,3]. Furthermore, OT can present to the ED in a variety of ways, making its diagnosis especially challenging as it can mimic various other gynecological and non-gynecological emergencies [4]. Differentiating ovarian torsion from other conditions with timely diagnosis and quick management are key to fertility preservation [5].

Although OT is a recognized gynecological emergency where timely ED diagnosis and management is critical to good outcomes, few studies have explored the key characteristics of OT cases presenting to an ED setting, their management practices within the ED context and the outcomes of these patients. Most studies that have explored the clinical and outcome aspect of OT presentations have combined OT cases from across different settings with the institution, where time to diagnosis, availability of imaging modalities and, ultimately outcomes, could vary [5–8]. Studies that have explored OT in the ED context have been limited in scope, with some focusing on specific populations (pediatric patients alone) [9–12] or on diagnostic modalities [2,13] or trend analysis [14]. One large retrospective analysis of the national US ED data looking at the utilization of the ED for OT, found an increase in the number of visits of OT in the decade reviewed, which nearly doubled from 1236 to 2695 [14]. Another study, looking at national database and types of gynecologic and obstetric emergencies in the pediatric and adolescent age group presenting to US EDs, found that while OT comprised less than 1% of obstetrical and gynecological presentations, it was associated with higher admission and transfer rates [10]. While these studies included large databases of ED visits, they did not review the clinical aspects of care of these patients within the ED context.

Our paper aims to highlight the incidence, risk factors, clinical presentation, physical examination findings, ED management practices, ultrasonographic and intraoperative findings, histopathology reports as well as the time-to-intervention metrics of OT cases presenting to the ED of a tertiary care center in Lebanon.

## Materials and methods

### Study setting and design

This study was conducted in the ED of a large tertiary care center at the American University of Beirut Medical Center (AUBMC), in Beirut, Lebanon. AUBMC's ED receives around 56,905 ED patient visits per year.

Approval for the present study was granted by the Institutional Review Board at AUBMC under the protocol ID BIO-2019-0015. We conducted a retrospective chart review of all patients of all age groups with OT who presented to the ED from January 1st, 2019 to December 31st, 2019. Data were accessed for research purposes from June, 2021 to December, 2021. All patients' identifiers were not accessed by authors during or after data collection. All patients with either ultrasound or intraoperative findings of OT were included in the study and their corresponding electronic charts were reviewed for data collection. Data elements included patient's risk factors, clinical presentation, physical examination, emergency management, ultrasonographic and intraoperative findings, histopathology reports as well as the time-

to-intervention metrics (door to ultrasonography, door to surgery and ED length of stay (LOS)) of OT cases presenting to the ED of our tertiary care center.

## Statistical analysis

The Statistical Package for Social Sciences (SPSS), version 24.0 was used for data entry and analyses. Descriptive analyses were carried out by calculating the number and percent for categorical variables, whereas the continuous variables were presented as the mean and standard deviation (±SD), or median and interquartile range (IQR), as appropriate.

## Results

A total of 20 confirmed cases of OT were included in the final review. The total number of women presenting to the ED over the study time period was 27,840. The total number of women of reproductive age (15–45 years old) over this time period was 12701. The incidence of OT was 71.8 per 100,000 ED visits of women. The incidence of OT in the ED was 157.4 per 100,000 visits of women in the reproductive age group.

Table 1 describes the 20 cases of OT including the duration of presenting symptoms, previous history of OT, pregnancy status, diagnostic modalities, ultrasonographic and intraoperative findings as well as the time-to-intervention metrics of OT cases presenting to the ED. A total of 15 patients (78.9%) presented within 24 hours of symptom onset, 1 patient (5.3%) presented within 48 hours and another 3 patients (15.8%) presented more than 72 hours of symptoms onset. Two patients (10%) were pregnant and two patients (10%) had a previous history of OT. A total of 16 patients (80%) underwent pelvic ultrasonography while 4 patients (20%) went straight to the operating room based on clinical suspicion. Amongst those who underwent pelvic ultrasonography, all had a color doppler examination done but only 13 cases had the doppler findings reported. Abnormal ovarian blood flow was detected in 8 cases (50%). A total of 12 (75%) exhibited ovarian cysts or masses, while 11 (68.9%) showed ovarian enlargement and ovarian edema was seen in 2 cases (12.5%). Peripheral displacement of follicles was noted in 6 cases (37.5%), and 5 cases (31.3%) showed abnormal ovarian location. The location of the OT whether by ultrasonography or intraoperatively was more on the right side rather than the left side with 60% and 40% respectively.

A total of 19 cases (95%) underwent surgery out of which (94.7%, 18/19) underwent ovarian detorsion and only 1 case (5.3%) had spontaneous detorsion intraoperatively. A single case (5.3%) did not undergo surgery although the ultrasonography done was highly suspicious of torsion but the decision to adopt a conservative management was deemed reasonable as patient became pain free. A total of 16 cases (84.2%) underwent cystectomy and 1 case had a cyst invading the fallopian tube and underwent salpingectomy. The mean number of times the ovary was torsed around its pedicle was noted to be 2.76 times with a range between 1 and 6 times. A normal appearing ovary intraoperatively was seen in 18 patients (94.7%), while 1 patient (5.3%), required salpingo-oophorectomy as the ovary remained necrotic post detorsion. A total of 13 cases included histopathologic review of specimens out of which 6 had mature cystic teratoma (46.2%), 3 had a hemorrhagic corpus luteal cyst (23.1%) and 2 had a benign serous cystadenoma (15.4%).

The mean time of door to ultrasonography was 1.4 hours, whereas the mean time from door to surgery was 11.4 hours. The mean ED LOS was found to be 3.6 hours.

Table 2 describes the baseline characteristics of the 20 patients with OT presenting to the ED. The mean age of the patients was 27.3 with an age range between 16 and 40. The mean BMI of all patients was 24.98 with a range between 21.37 and 29.05. A total of 5 patients (25%) had a history of polycystic ovarian syndrome (PCOS), 3 patients (15%) underwent a recent In-

Table 1. Case description of the 20 ovarian torsion cases along with the ED management data.

| Case | Age | Duration of presenting symptoms | Previous episodes of OT | Pregnancy | Diagnostic modality | Doppler ultrasonography findings | Intraoperative findings | Intraoperative procedure | Histopathology report | Door to ultrasonography | Door to surgery | ED LOS |
|---|---|---|---|---|---|---|---|---|---|---|---|---|
| 1 | 31 | 1 hour | No | No | US | • Enlarged + edematous right ovary • Large cyst | • Right OT • Viable ovary | • Detorsion + Cystectomy | • Follicular cysts • Hemorrhagic corpus luteal cyst. | 2hr | 19hr 40min | 4hr 35min |
| 2 | 26 | 48 hours | No | No | US | • Heterogeneous right pelvic mass • Doppler: No detectable arterial flow | • Right OT x 1 time • Viable ovary | • Detorsion + Cystectomy | • Mature cystic teratoma | 4hr 30min | 5hr 10min | 7hr 50min |
| 3 | 31 | 1 hour | No | No | US | • Right Adnexal/Para ovarian cyst. • Doppler: No detectable arterial flow. | • Right OT x 3 times • Viable ovary | • Detorsion + Cystectomy | • Benign serous cystadenoma | 35min | 4hr 40min | 4hr 20min |
| 4 | 27 | 3 hours | No | No | Clinical | N/A | • Left OT x 3 times • Viable ovary | • Detorsion + cystectomy | • Mature cystic teratoma | N/A | 12hr 30min | 6hr 10min |
| 5 | 29 | <24 hours | No | No | US | • -Markedly enlarged left ovary • Simple cyst | • Left OT x 3 times • Viable ovary | • Detorsion + Cystectomy | N/A | 2hr 40min | 7hr 30min | 4hr 10min |
| 6 | 35 | 72 hours | No | No | Clinical | N/A | • Right OT x 2 times • Viable ovary | • Detorsion + Cystectomy | • Mature cystic teratoma | N/A | 4hr 50min | 4hr 20min |
| 7 | 16 | <24 hours | No | No | US | • High position of left ovary in the pelvis • Large left ovarian cyst • Multiple peripherally distributed follicles • Doppler: Normal flow | • Left OT x 2 times • Viable ovary | • Detorsion + Cystectomy + Salpingectomy | • Benign serous cystadenoma • Benign fallopian tube | 30 min | 4hr 50min | 4hr 20min |
| 8 | 28 | 72 hours | No | No | US | • Markedly enlarged left ovary • Peripherally displaced follicles • Doppler: No detectable arterial flow | • Left OT x 6 times • Non-viable ovary | • Detorsion attempt → ovary remain bluish • Salpingo-oophorectomy | • Fallopian tube and ovarian tissue with hemorrhage and vascular congestion | 3hr | 5hr 20min | 20min |
| 9 | 24 | 168 hours | No | No | US | • Markedly enlarged right ovary • Large ovarian cyst • Peripherally displaced follicles • Doppler: No detectable arterial flow | • Ovary detorsed spontaneously • Viable ovary | • Cystectomy | • Hemorrhagic corpus luteal cyst • Large caliber vessel noted | 1hr 15min | 4hr 25min | 3hr 45min |

(Continued)

**Table 1.** (Continued)

| Case | Age | Duration of presenting symptoms | Previous episodes of OT | Pregnancy | Diagnostic modality | Doppler ultrasonography findings | Intraoperative findings | Intraoperative procedure | Histopathology report | Door to ultrasonography | Door to surgery | ED LOS |
|---|---|---|---|---|---|---|---|---|---|---|---|---|
| 10 | 38 | 24 hours | No | No | Clinical | N/A | • Right OT x 6 times<br>• Viable ovary | • Detorsion + Cystectomy | • Mature cystic teratoma | N/A | 5hr | 2hr 30min |
| 11 | 18 | 24 hours | No | No | US | • Enlarged left ovary<br>• Heterogeneous cysts<br>• Doppler: No detectable arterial flow<br>• Findings suggestive of acute ovarian torsion | • Left OT x 2 times<br>• Viable ovary | • Detorsion + Cystectomy | N/A | 2hr 40min | 8hr 45min | 3hr 30min |
| 12 | 34 | 2 hours | No | Yes | Clinical | N/A | • Left OT x 2 times<br>• Viable ovary | • Detorsion + Cystectomy | • Hemorrhagic corpus luteum cyst | N/A | 13hr 20min | 1hr 10min |
| 13 | 20 | 24 hours | No | No | US | Right ovary could not be visualized with certainty | • Right OT x 4 times<br>• Viable ovary | • Detorsion + Cystectomy | • Mature cystic teratoma | 2hr | 8hr 40min | 5hr |
| 14 | 33 | 24 hours | No | No | US | • Right ovarian teratoma<br>• Doppler: Normal flow | • Right OT x 3 times<br>• Viable ovary | • Detorsion + Cystectomy | • Mature cystic teratoma | Done on the second day (not in ED) | 3hr 22min | 2hr 45min |
| 15 | 16 | 2 hours | No | No | US | • Markedly enlarged + Medially slightly superiorly displaced left ovary<br>• Heterogeneous and hyperechoic parenchyma<br>• Doppler: Normal flow | • Left OT x 1 time<br>• Viable ovary | • Detorsion | N/A | 25 min | 4hr 52min | 4hr 10 min |
| 16 | 24 | 24 hours | yes | No | US | • Markedly enlarged right ovary + displaced superiorly<br>• Peripherally displaced follicles<br>• Large cystic lesion<br>• Doppler: No detectable arterial flow | • Right OT x 3 times<br>• Viable ovary | Detorsion + Cystectomy | • Benign hemorrhagic cyst | 4 min | 15hr 45min | 1hr 20min |
| 17 | 40 | 1 hour | No | No | US | • Markedly enlarged + edematous left ovary<br>• Peripherally displaced follicles<br>• Doppler: Normal flow | • Left OT x 1.5 times<br>• Viable ovary | • Detorsion + Cystectomy | N/A | 7 min | 9hr 20min | 3hr 45min |

(*Continued*)

**Table 1.** (Continued)

| Case | Age | Duration of presenting symptoms | Previous episodes of OT | Pregnancy | Diagnostic modality | Doppler ultrasonography findings | Intraoperative findings | Intraoperative procedure | Histopathology report | Door to ultrasonography | Door to surgery | ED LOS |
|---|---|---|---|---|---|---|---|---|---|---|---|---|
| **18** | 24 | 1 hour | yes | No | US | • Markedly enlarged right ovary<br>• Peripherally displaced follicles<br>• Doppler: No detectable arterial flow | • Right OT x 2.5 times<br>• Viable ovary | • Detorsion | N/A | 10 min | 3hr | 1hr 20min |
| **19** | 27 | 1 hour | No | Yes | US | • Multiple large ovarian cysts bilaterally<br>• -Enlarged right ovary + displaced superiorly<br>• Doppler: normal flow | N/A | No surgery done–Conservative Management | N/A | 1hr | N/A | 4hr 10min |
| **20** | 25 | NA | No | No | US | • Markedly enlarged right ovary + displaced superiorly and to the midline<br>• Multiple large simple cysts and least one complicated/hemorrhagic cyst<br>• Doppler: No detectable arterial flow | • Right OT x 2 times<br>• Viable ovary | • Detorsion + Cystectomy | N/A | 13 min | 2hr 35min | 2hr 15min |

**Table 2. Baseline characteristics of patients with OT presenting to the emergency department.**

|  | Total N = 20 |
|---|---|
| **Age (years)** Mean (SD) (Min-Max) | 27.3 (6.8) (16–40) |
| **BMI** Mean (SD) (Min-Max) | 24.98 (2.61) (21.37–29.05) |
| **Past Medical History** |  |
| PCOS or ovarian cysts | 5 (25%) |
| IVF | 3 (15%) |
| **Smoker** | 7 (35%) |
| **Alcohol** | 2 (10%) |
| **Home Medications** |  |
| Oral contraceptive pills/Progesterone | 2 (10%) |

Vitro-Fertilization (IVF) treatment/procedure. Only 7 patients (35%) were smokers and 2 patients (10%) consumed alcohol. A total of 2 patients (10%) were on oral contraceptive pills/ progesterone.

Table 3 describes the clinical presentation and physical examination of patients with OT in the ED.

Nineteen patients (95%) presented with abdominal pain out of which 16 (75%) reported the pain to be severe in nature and 12 (60%) reported the pain to be localized to the right lower quadrant. A total of 18 patients (90%) reported nausea and vomiting. Some patients presented with vaginal bleeding, fever, chills, back pain as well as flank and pelvic pain. As for the physical examination, abdominal tenderness was the most common physical exam finding (65%). The most common location of the abdominal tenderness was in the right lower quadrant (45%). Some patients also presented with abdominal rigidity, guarding and rebound tenderness associated with peritoneal irritation.

Table 4 describes the medication administered to the patients during their stay in the ED. In fact, 19 patients (95%) received opioids, 18 patients (90%) received acetaminophen, and 80% received NSAID for pain control. As antiemetics, 18 patients (90%) received either Ondansetron and/or Metoclopramide. Esomeprazole was given to 12 patients (60%).

Table 5 describes the types of cysts detected by pelvic ultrasonography of patients with OT. A total of 12 patients had ovarian cyst of various types: 5 cases (31.3%) of dermoid cysts, 4 cases (25%) of simple cysts, 1 case (6.25%) of mass teratoma, 1 case (6.25%) of a hemorrhagic cyst, and 1 case (6.25%) of a paraovarian cyst.

## Discussion

This study describes the incidence, risk factors, clinical presentation, physical examination, emergency management, ultrasonographic and intraoperative findings, histopathology reports as well as the time-to-intervention metrics of OT cases presenting to the ED of our tertiary care center. A total of 20 confirmed cases of OT over a period of 1 year were included in our study. The incidence of OT in the ED was 157.4 per 100,000 visits of women in the reproductive age group. A minority of our patients had a history of prior OT, were pregnant, had a previous diagnosis of PCOS or were on oral contraceptive pills. The majority of our patients presented to the ED within 24 hours of symptoms onset, had abdominal pain localizing to the right lower quadrant and required opioid for pain and antiemetics for nausea. Moreover, the majority of our patients were diagnosed by pelvic ultrasonography prior to surgical diagnosis, with only four proceeding directly to the operating theater based on clinical suspicion alone. Ovarian cyst, with ovarian enlargement and abnormal ovarian blood flow were the most

**Table 3. Clinical presentation and physical examination of patients with OT in the emergency department.**

| Symptoms | |
|---|---|
| Abdominal Pain | 19 (95%) |
| Nausea or vomiting | 18 (90%) |
| Vaginal bleeding | 1 (5%) |
| Fever | 1 (5%) |
| Chills | 1 (5%) |
| Back pain | 1 (5%) |
| Flank pain | 2 (10%) |
| Pelvic Pain | 1 (5%) |
| **Severity of Symptoms** | |
| Severe | 16 (75%) * |
| **ESI (Emergency Severity Index)** | |
| 2 | 1 (5%) |
| 3 | 19 (95%) |
| **Location of Abdominal Pain** | |
| Right Lower pain | 12 (60%) |
| Left Lower pain | 3 (15%) |
| Right Flank pain | 2 (10%) |
| Left Flank pain | 3 (15%) |
| Back | 1 (5%) |
| Suprapubic | 1 (5%) |
| **Physical Examination** | |
| Abdominal Tenderness | 13 (65%) |
| Abdominal Rigidity | 2 (10%) |
| Guarding | 3 (15%) |
| Rebound tenderness | 3 (15%) |
| **Location of Abdominal Tenderness** | |
| Right upper quadrant | 1 (5%) |
| Left upper quadrant | 2 (10%) |
| Right lower quadrant | 9 (45%) |
| Left lower quadrant | 4 (20%) |
| Right Flank | 3 (15%) |
| Left Flank | 2 (10%) |
| Suprapubic | 3 (15%) |

* 4 missing

**Table 4. ED medications given to patients with OT.**

| | Total N = 20 |
|---|---|
| **Medication given in ED** | |
| Opioids | 19 (95%) |
| Paracetamol | 18 (90%) |
| Antiemetics (Ondansetron/Metoclopramide) | 18 (90%) |
| NSAID | 16 (80%) |
| Esomeprazole | 12 (60%) |

**Table 5. Ultrasonographic findings of patient with OT.**

|  | Total N = 16 |
|---|---|
| **Ultrasound detected cysts** | |
| Dermoid cyst | 5 (31.3%) |
| Simple cyst | 4 (25%) |
| Mass Teratoma | 1 (6.25%) |
| Hemorrhagic cyst | 1 (6.25%) |
| Para ovarian cyst | 1 (6.25%) |

common ultrasonographic findings observed. Most patients underwent operative management with viable ovaries post ovarian detorsion with the exception of one patient who presented 72 hours after symptom onset. The longest duration of symptoms with viable ovary intraoperatively was 168 hours. The most common finding on histopathologic review was mature cystic teratoma.

To our knowledge, our study is one of the few studies in the literature to comprehensively describe OT clinical and outcome variables from an emergency medicine perspective. Studies that have described the clinical and outcome of patients with OT have combined cases from across different settings (ambulatory and emergency department) which could impact the diagnostic modalities used, time-to intervention metrics and outcome of patients. ED specific studies on the other hand have had more limited scopes with some focusing on pediatric patients alone, others looking at diagnostic modalities and others looking at trend analysis [2,6,10,14]. This study is unique in comprehensively reviewing all aspects of clinical care and outcome metrics of both adult and pediatric OT cases in the ED setting.

OT is considered a rare condition accounting for around 2.7% of gynecological surgical emergencies [3], and around 3% of ED visits presenting with abdominal discomfort [5]. Few studies have investigated the incidence of OT in the ED. In our study, the incidence of OT was 71.8 per 100,000 ED visits of women of all ages, and 157.4 per 100,000 ED visits of women of reproductive age (15–45 years old). While there is no ED incidence comparative data, a study in Korea looking at incidence amongst inpatient admissions reported a rate 5.9 and 9.9 per 100,000 inpatient visits of women all ages and women of reproductive age respectively [15]. We believe that the incidence of OT is higher in our population where there is a high prevalence of PCOS, a known risk factors for OT [16,17]. In our study having a history of PCOS was only reported in 25% of the patients, which is similar to a 5 year retrospective case series in India in which PCOS was found in 25.7% of patients with OT [18]. In fact, the age-standardized rate of PCOS in Lebanon in 2019 was 89.7 cases per 100,000 women which is higher than most countries in the MENA and the Middle East with an age-standardized rate of 77.2 cases per 100 000 women [16] and in central and eastern Europe with an age-standardized rate of 74 cases per 100 000 women in 2017 [19]. Moreover, in our study, the most common histopathologic finding reported was mature cystic teratoma (46.2%) in comparison to a study done in India in which mature cystic teratoma accounted for only 16.7% [20]. Mature cystic teratomas share similar pathophysiologic patterns of hyperandrogenism with PCOS. Further studies are needed to explore the pathophysiologic characteristics of our patient population that may be contributing to our findings.

All cases of OT in our study were in women of reproductive age. Indeed, most cases of OT occurs in women of reproductive age and is usually less common in premenarchal girls and postmenopausal women [21]. Two of our patients (10%) were pregnant and 3 of our patients (15%) underwent a recent IVF procedure. Indeed, about 10%-22% of OT occurs in pregnancy [7,18,22], and can occur in about 12–20% of women undergoing IVF with ovarian

hyperstimulation syndrome [23,24]. Furthermore, 2 of our patients (10%) had a previous history of OT. Indeed, patients with a history of OT might be at increased risk of recurrence [25]. One series found that 23 out of 216 cases (11%) were recurrent [26].

The diagnosis of OT in the ED remains a challenge for the emergency physician given the broad differential of patients presenting with acute abdominal pain. In our study, we found that the majority of our patients presented within 24 hours of symptoms onset, with 95% of them having abdominal pain, 75% reporting the pain to be severe in nature, 60% describing the pain to be localized to the right lower quadrant and 90% reporting nausea and vomiting. Our study aligns with other similar studies in the literature that also found that there is a higher rate of right-sided adnexal torsion (64%) and pain to be reported as severe (69%). The most commonly reported duration of pain was also 24 hours. Nausea is also a commonly reported symptom present in the majority of our patients (90%) similar to findings in the literature in which nausea/vomiting ranged between 67 and 83% [11]. Moreover, in our study we have found that 95% of women required opioids and 90% required antiemetics. This is reflective of the severity of pain that is common in OT patients.

Although, the gold standard for the diagnosis of OT remains by laparoscopic surgical exploration, pelvic ultrasonography with color doppler is the diagnostic imaging of choice [5,27]. In fact, the sensitivity of ultrasonography to detect OT was found to range from 70%-72% and the specificity from 87%-99.6% [27,28]. In our study, 80% of our patients underwent pelvic ultrasonography with ovarian cysts or masses, ovarian enlargement, peripheral displacement of follicles, abnormal ovarian location and abnormal ovarian blood flow being the most common ultrasonographic findings noted in our OT patients. Indeed, similar findings were noted in the literature with the exception of abnormal ovarian blood flow which did not yield any statistical significance in being an ultrasonography finding characteristic of OT [27,29,30].

Indeed, given the above findings, one should not rely solely on pelvic ultrasonography as it can sometime miss a diagnosis of OT. Point of care ultrasonography should be used as an adjunct to the clinical presentation and physical examination while excluding other non-gynecological emergencies that can mimic the presentation of OT. In fact, in our study 20% of patients went to the operating theater based on clinical suspicion alone before OT was confirmed intraoperatively. Some studies have tried to identify a predictive score of a pre-operative diagnosis of OT. In fact, in a prospective cohort study of 35 women with acute pelvic pain found that patients presenting with unilateral abdominal pain, with a duration of less than 8 hours at first presentation, vomiting, absence of leucorrhoea and metrorrhagia, and ovarian cyst larger than 5 cm by ultrasonography, have a higher likelihood of having an intraoperative diagnosis of OT [13].

Moreover, in our study, almost all of our patients underwent laparoscopic ovarian detorsion out of which 94.7% had viable ovaries with only 1 patient requiring salpingo-oophorectomy as the ovary remained necrotic post detorsion. This patient presented to the ED after 72 hours of symptom onset. Most of our patients presented within 24 hours of symptoms onset and the mean time from door to surgery was 11.4 hours. Indeed, the current surgical best practices is oriented towards ovarian salvageability with fertility preservation techniques post laparoscopic ovarian detorsion [31]. Our single patient who required salpingo-oophorectomy presented at 72 hours of symptoms onset, had the surgery done at 5 hours and 20 minutes and had the ovary rotated 6 times around its pedicle. However, a second patient who presented to our emergency room within 24 hours of symptoms onset, was operated on at 5 hours from presentation and who had also the ovary rotated 6 times around its pedicle had a viable ovary post detorsion. A study found that the median duration from the onset of pain symptoms to presentation (26.0 vs 6.0 h, p < .001) and from presentation to surgery (11.0 vs 5.5 h, p = .010) were significantly longer in women who required an oophorectomy compared to women who

had conservative surgery [6]. Indeed, a study looking at the duration of symptoms onset and ovarian salvageability found that the mean duration of symptoms prior to surgery was 87 hours with the longest duration of symptoms of 159 hours. In our study one patient had symptoms lasting for 168 hours and had viable ovary post ovarian detorsion. Indeed, our results align with the current literature that although the duration from symptoms onset to presentation plays an important role in understanding ovarian salvageability, multiple other factors should be considered such as patient's age, diagnostic urgency and prompt surgical intervention [6,32].

Our study has several limitations, the most pertinent of which is the small sample size, as it is a single centered retrospective chart review over a short period of 1 year, thus limiting the generalizability of the findings. However, this represented a sample from the largest ED in Lebanon. Given the retrospective design of this study, missing data constitute additional limitations as most of our data were collected from medical charts and physician notes.

Our study highlights the 20 cases of women with OT presenting to the emergency department of our tertiary care center over a period of 1 year. It is one of the very few studies that comprehensively addressed the topic of OT specifically from an emergency medicine perspective by focusing on the incidence, risk factors, clinical presentations, physical examination, emergency management, ultrasonographic and intraoperative findings, histopathology reports as well as time-to-intervention metrics of OT cases. We have found that we have a higher incidence of OT in our population compared to the literature. Pregnancy, IVF treatment and history of previous OT were identified as common factors in our population. Severe abdominal pain and localization to the right lower quadrant are common clinical presentation as are nausea and vomiting. Ultrasonography remains the most commonly used radiographic modality, however, intraoperative diagnosis remains the gold standard particularly for patients with a high clinical suspicion. Finally, ovaries can remain salvageable even days after onset of symptoms.

Future ED studies designed to explore clinical and diagnostic predictors of OT may help with prompt timely diagnosis and management of these patients.

## Supporting information

**S1 Data. The minimal data set underlying the results described in the manuscript.** (XLSX)

## Author Contributions

**Conceptualization:** Eveline Hitti.

**Data curation:** Faysal Tabbara.

**Formal analysis:** Faysal Tabbara.

**Investigation:** Faysal Tabbara, Moustafa Hariri, Eveline Hitti.

**Methodology:** Faysal Tabbara, Moustafa Hariri.

**Project administration:** Moustafa Hariri, Eveline Hitti.

**Supervision:** Eveline Hitti.

**Writing – original draft:** Faysal Tabbara, Moustafa Hariri, Eveline Hitti.

**Writing – review & editing:** Faysal Tabbara, Moustafa Hariri, Eveline Hitti.

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
