## [Decision Letter · Decision Letter 0]

15 Dec 2023

PONE-D-23-28017Ovarian torsion: A retrospective case series at a tertiary care center emergency departmentPLOS ONE

Dear Dr. Hitti,

Thank you for submitting your manuscript to PLOS ONE. After careful consideration, we feel that it has merit but does not fully meet PLOS ONE’s publication criteria as it currently stands. Therefore, we invite you to submit a revised version of the manuscript that addresses the points raised during the review process. Please, kindly upload the histopathology report of the operated case and clarify whether Doppler was performed in all cases along with USG. Please submit your revised manuscript by Jan 28 2024 11:59PM. If you will need more time than this to complete your revisions, please reply to this message or contact the journal office at plosone@plos.org. Please include the following items when submitting your revised manuscript:A rebuttal letter that responds to each point raised by the academic editor and reviewer(s). You should upload this letter as a separate file labeled 'Response to Reviewers'.A marked-up copy of your manuscript that highlights changes made to the original version. You should upload this as a separate file labeled 'Revised Manuscript with Track Changes'.An unmarked version of your revised paper without tracked changes. You should upload this as a separate file labeled 'Manuscript'.If applicable, we recommend that you deposit your laboratory protocols in protocols.io to enhance the reproducibility of your results. Protocols.io assigns your protocol its own identifier (DOI) so that it can be cited independently in the future. For instructions see: https://journals.plos.org/plosone/s/submission-guidelines#loc-laboratory-protocols. Additionally, PLOS ONE offers an option for publishing peer-reviewed Lab Protocol articles, which describe protocols hosted on protocols.io. Read more information on sharing protocols at https://plos.org/protocols?utm_medium=editorial-email&utm_source=authorletters&utm_campaign=protocols.

We look forward to receiving your revised manuscript.

Kind regards,

Fabio Vasconcellos Comim, MD,PhD

Academic Editor

PLOS ONE

Journal Requirements:

Did you know that depositing data in a repository is associated with up to a 25% citation advantage (https://doi.org/10.1371/journal.pone.0230416)? If you’ve not already done so, consider depositing your raw data in a repository to ensure your work is read, appreciated and cited by the largest possible audience. You’ll also earn an Accessible Data icon on your published paper if you deposit your data in any participating repository (https://plos.org/open-science/open-data/#accessible-data).

Reviewers' comments:

Reviewer's Responses to Questions

**Comments to the Author**

1. Is the manuscript technically sound, and do the data support the conclusions?

Reviewer #1: Yes

Reviewer #2: Yes

2. Has the statistical analysis been performed appropriately and rigorously? 

Reviewer #1: Yes

Reviewer #2: Yes

3. Have the authors made all data underlying the findings in their manuscript fully available?

Reviewer #1: Yes

Reviewer #2: Yes

4. Is the manuscript presented in an intelligible fashion and written in standard English?

Reviewer #1: Yes

Reviewer #2: Yes

5. Review Comments to the Author

Reviewer #1: The topic is important and not rare at all. Although the final number of cases was twenty cases only but I know it is not ease to find a large number of cases with evident torsion especially in one hospital.

Reviewer #2: Dear author, the study is good. Kindly upload the histopathology report of the operated case. Was Doppler done in all cases along with USG. In how many cases the whirlpool sign was positive

6. PLOS authors have the option to publish the peer review history of their article (what does this mean?). If published, this will include your full peer review and any attached files.

Reviewer #1: No

Reviewer #2: **Yes: **Prof Dr Varsha Deshmukh

---

## [Author Response · Author response to Decision Letter 0]

27 Dec 2023

Rebuttal Letter that responds to each point raised by the academic editor and reviewers

Dear Drs. Fabio Vasconcellos, Emily Chenette and Iain Hrynaszkiewicz, 

Thank you very much for your positive response and for considering our manuscript for publication at your esteemed journal. We are submitting a revised version of the manuscript after addressing the minor comments as per the academic editor and reviewers. 

Thank you for your comment. We ensured that the manuscript meets the PLOS ONE’s style requirements. 

Did you know that depositing data in a repository is associated with up to a 25% citation advantage (https://doi.org/10.1371/journal.pone.0230416)? If you’ve not already done so, consider depositing your raw data in a repository to ensure your work is read, appreciated and cited by the largest possible audience. You’ll also earn an Accessible Data icon on your published paper if you deposit your data in any participating repository (https://plos.org/open-science/open-data/#accessible-data).

Thank you for your note. We appreciate you informing us about depositing the raw data in a repository. 

Thank you for your comment. All our data is fully available without any restrictions. We totally agree of the importance of sharing our data used to reach the conclusions drawn and to replicate the study findings in their entirety. We uploaded the study’s minimal underlying data set as a supporting information file. 

Thank you for your comment. We have reviewed the reference list and ensured that it is complete and correct. We have added 1 additional reference in the discussion section and in the reference list to highlight the histopathologic findings noted in our study as compared to the literature. 

20. Vijayalakshmi K. Clinico-Pathological Profile of Adnexal Torsion Cases: A Retrospective Analysis from A Tertiary Care Teaching Hospital. J Clin Diagn Res [Internet]. 2014 [cited 2023 Dec 22]; Available from: http://jcdr.net/article_fulltext.asp?issn=0973-709x&year=2014&volume=8&issue=6&page=OC04&issn=0973-709x&id=4456

Comments to the Author

1. Is the manuscript technically sound, and do the data support the conclusions?

Reviewer #1: Yes

Reviewer #2: Yes

Thank you for your comment. 

2. Has the statistical analysis been performed appropriately and rigorously?

Reviewer #1: Yes

Reviewer #2: Yes

Thank you for your comment. 

3. Have the authors made all data underlying the findings in their manuscript fully available?

Reviewer #1: Yes

Reviewer #2: Yes

Thank you for your comment. All data underlying the findings described in the manuscript is fully available without restrictions. We uploaded the study’s minimal underlying data set as a supporting information file.

4. Is the manuscript presented in an intelligible fashion and written in standard English?

Reviewer #1: Yes

Reviewer #2: Yes

Thank you for your comment. 

5. Review Comments to the Author

Reviewer #1: The topic is important and not rare at all. Although the final number of cases was twenty cases only but I know it is not ease to find a large number of cases with evident torsion especially in one hospital.

Thank you for your comment. Indeed, while the overall number of cases is not large, this incidence in one setting was and we highlighted this in our discussion section. 

Reviewer #2: Dear author, the study is good. Kindly upload the histopathology report of the operated case. Was Doppler done in all cases along with USG. In how many cases the whirlpool sign was positive.

Thank you for your constructive suggestions. We uploaded the histopathology results of the operated cases in table 1, in a new column under the heading: “Histopathology report”. A total of 13 cases have a histopathology result. 7 cases do not have any histopathology results (marked as N/A in the table) given that either no specimen was taken during surgery or because of missing data. We have updated these results in the manuscript. We also noted that the most common histopathology finding was mature cystic teratoma. We discussed these findings in our discussion section. 

Moreover, while reviewing the histopathology reports we have identified 5 additional cases of cystectomies, that were not previously noted in the US reports/operative findings. We have added these cases and modified these findings and percentages in the entire manuscript. 

While all of the cases who underwent pelvic ultrasonography had a color doppler examination done, only 13 of the reported interpretations commented on the doppler findings. We have documented these doppler findings in table 1 under the heading: “Doppler ultrasonography findings” and subheading “Doppler” when available. In addition, even though the whirlpool sign has a high sensitivity and specificity in the diagnosis of ovarian torsion, it was not documented as a terminology in any of the ultrasonographic reports by the radiology team at our tertiary care center. 

6. PLOS authors have the option to publish the peer review history of their article (what does this mean?). If published, this will include your full peer review and any attached files.

Do you want your identity to be public for this peer review? For information about this choice, including consent withdrawal, please see our Privacy Policy.

Reviewer #1: No

Reviewer #2: Yes: Prof Dr Varsha Deshmukh

Thank you for your comment.

---

## [Editor Report · Decision Letter 1]

11 Jan 2024

Ovarian torsion: A retrospective case series at a tertiary care center emergency department

PONE-D-23-28017R1

Dear Dr. Hittl,

We’re pleased to inform you that your manuscript has been judged scientifically suitable for publication and will be formally accepted for publication once it meets all outstanding technical requirements.

Kind regards,

Fabio Vasconcellos Comim, MD,PhD

Academic Editor

PLOS ONE

---

## [Editor Report · Acceptance letter]

26 Feb 2024

PONE-D-23-28017R1 

PLOS ONE

Dear Dr. Hitti, 

I'm pleased to inform you that your manuscript has been deemed suitable for publication in PLOS ONE. Congratulations! Your manuscript is now being handed over to our production team.

Kind regards, 

on behalf of

Prof Fabio Vasconcellos Comim 

Academic Editor

PLOS ONE